# Sulforaphane Elicits Protective Effects in Intestinal Ischemia Reperfusion Injury

**DOI:** 10.3390/ijms21155189

**Published:** 2020-07-22

**Authors:** Zhiquan Chen, Annika Mohr, Barbara Heitplatz, Uwe Hansen, Andreas Pascher, Jens G. Brockmann, Felix Becker

**Affiliations:** 1Department of General, Visceral and Transplant Surgery, University Hospital Münster, 48149 Münster, Germany; chen_zhiquan@outlook.com (Z.C.); andreas.pascher@ukmuenster.de (A.P.); jens.brockmann@ukmuenster.de (J.G.B.); felix.becker@ukmuenster.de (F.B.); 2Gerhard Domagk Institute of Pathology, University Hospital Muenster, 48149 Münster, Germany; Barbara.heitplatz@ukmuenster.de; 3Department of Molecular Medicine, Institute for Musculoskeletal Medicine, University Hospital Muenster, 48149 Münster, Germany; uwe.hansen@ukmuenster.de

**Keywords:** sulforaphane, intestinal reperfusion injury

## Abstract

Intestinal ischemia reperfusion injury (IRI) is an inherent, unavoidable event of intestinal transplantation, contributing to allograft failure and rejection. The inflammatory state elicited by intestinal IRI is characterized by heightened leukocyte recruitment to the gut, which is amplified by a cross-talk with platelets at the endothelial border. Sulforaphane (SFN), a naturally occurring isothiocyanate, exhibits anti-inflammatory characteristics and has been shown to reduce platelet activation and block leukocyte adhesion. Thus, the aim of this study was to investigate protective effects and mechanism of action of SFN in a murine model of intestinal IRI. Intestinal IRI was induced by superior mesenteric artery occlusion for 30 min, followed by reperfusion for 2 h, 8 h or 24 h. To investigate cellular interactions, leukocytes were in vivo stained with rhodamine and platelets were harvested from donor animals and ex vivo stained. Mice (C57BL/6J) were divided into three groups: (1) control, (2) SFN treatment 24 h prior to reperfusion and (3) SFN treatment 24 h prior to platelet donation. Leukocyte and platelet recruitment was analyzed via intravital microscopy. Tissue was analyzed for morphological alterations in intestinal mucosa, barrier permeability, and leukocyte infiltration. Leukocyte rolling and adhesion was significantly reduced 2 h and 8 h after reperfusion. Mice receiving SFN treated platelets exhibited significantly decreased leukocyte and platelet recruitment. SFN showed protection for intestinal tissue with less damage observed in histopathological and ultrastructural evaluation. In summary, the data presented provide evidence for SFN as a potential therapeutic strategy against intestinal IRI.

## 1. Introduction

Intestinal ischemia reperfusion injury (IRI) is an inherent, unavoidable event of intestinal transplantation, critically contributing to high rates of intestinal allograft failure and rejection [1]. The hallmark event of intestinal IRI is a characteristic inflammatory state, elicited, by heightened leukocyte recruitment into the gut, which is amplified by a cross-talk with platelets at the endothelial border [1,2]. While this sequence is inherent to intestinal transplantation, intestinal IRI is also observed in a variety of diseases ranging from visceral arterial occlusion, mesenteric venous thrombosis and nonocclusive low-flow situations such as cardiac insufficiency and hypovolemia [3,4]. With the resumption of blood flow at reperfusion, damage associated molecular patterns (DAMPs) interact with pattern recognition receptors (PRRs) such as Toll-like receptors (TLRs) on immune as well as vascular endothelial cells (VEC), triggering activation of downstream signaling [2,5]. This activation leads to production of proinflammatory cytokines and chemokines as well as subsequent endothelial activation [2]. All of this further corroborates the immunological recognition of intestinal graft (antigenicity) and elicits leucocytes from the systemic circulation to be recruited to the inflamed site and cause further pathological injury.

Leukocyte recruitment and infiltration is involving a sequence of events, including selectin-mediated rolling, chemokine-mediated activation, integrin-mediated firm adhesion and ultimately transmigration [6]. Interaction between leukocytes and VEC depends on expression of adhesion molecules such as P-selectin, E-selectin, intracellular adhesion molecule-1 (ICAM-1) and vascular cell adhesion molecule-1 (VCAM-1), subsequently leading to intercellular signaling pathway regulation [7,8,9]. Alongside with leukocytes, platelets are more and more revealed playing a critical role in the process of IRI [7,10,11]. Platelets are activated early after reperfusion and are among the first cells recruited to the site of injury [12]. Activated platelets not only secrete proinflammatory molecules, but also adhere to endothelial cells and leukocytes and support leukocyte transmigration into inflamed site [7,9,10,11,13]. Leukocyte capture is mediated by platelet P-selectin binding to leukocyte P-selectin glycoprotein ligand-1 (PSGL-1) to establish heterotypic adhesion. Hence, activation of platelets during reperfusion of ischemic tissue and their recruitment of inflammatory leukocytes significantly affects the severity of reperfusion injury.

The sulfur-based isothiocyanate L-sulforaphane (SFN, (*R*)-1-isothiocyanato-4-(methylsulfinyl) butane, 4-methylsulfinylbutyl isothiocyanate, C_6_H_11_NOS_2_) is a naturally occurring isothiocyanate found in cruciferous vegetables [14]. SFN has been demonstrated to induce anti-oxidative capacities and protect against cell inflammatory stress via interaction with the NF-erythroid 2 related factor 2 (Nrf2) transcription factor [15,16]. Nrf2 then binds to the antioxidant response element (ARE) in the promoter region of several genes accountable for antioxidant protection including heme oxygenase-1 (HO-1), NAD(P)H: quinoine oxidoreductase 1, glutathione reductase, and glutathione peroxidase (GSH-Px) [17,18,19]. Especially, the encoded phase 2 enzymes play a major role in the detoxification of ROS during IRI [15]. Moreover, it has been shown that SFN can affect cells of the innate immune system including neutrophils and platelets that are known as first line defense following ischemic injury. SFN-pretreatment not only influences platelets in terms of aggregation and activation, but also alters leukocyte adhesion and leukocyte-endothelial interaction, thus, potentially ameliorating inflammatory state by reducing leukocyte adhesion. [20,21,22].

Given the known key role of leukocyte adhesion in the pathophysiology of intestinal IRI and the established pathogenic role of platelets in the initiation and perpetuation of intestinal inflammation by enhancing leukocyte adhesion, aim of this study was to test if SFN is protective in intestinal IRI and if this protection is in-part mediated via an effect on platelets which leads to decreased leukocyte adhesion. Hence, this study was conducted to (a) characterize the protective effects of SFN treatment in a murine model of intestinal IRI with a specific focus on leukocyte adhesion and to (b) investigate into whether intestinal IRI-induced leukocyte recruitment can be reduced by pretreatment of platelets with SFN.

## 2. Results

### 2.1. SFN Treated Mice were Less Susceptible to Intestinal IRI

To test whether SFN could improve IRI-induced morphological damage in the murine intestine, H&E stained cross sections were evaluated and graded according to the Park/Chiu score. After 2 h reperfusion control animals displayed a mixed inflammatory cell infiltrate in the submucosa as well as pronounced crypt layer injury (Figure 1A). In SFN-treated animals, histological damage was markedly ameliorated and especially disintegration of the lamina propria and denudation of the epithelium at the villi were reduced (Figure 1B, insert). This was statistically significant when calculating the numeric Park/Chiu score, comparing control and SFN-treated animals after 2 h reperfusion (Figure 1G). After 8 h reperfusion, histological damage was already decreasing, however dilated capillaries as well as detached epithelial cells at the villus tips were still present in control animals (Figure 1C, insert).

In comparison, SFN-treated animals showed only slight signs of mucosal injury (Figure 1D). Again, differences were statistically significant (Figure 1G). After 24 h reperfusion, control animals exhibited a further decreased injury and only slight lifting at the villi sides (Figure 1E, insert), while SFN-treated animals showed a nearly normal mucosal anatomy (Figure 1F). Although, the Park/Chiu score after 24 h reperfusion for control animals was still higher compared to SFN-treated animals, this did not reach statistical significance (Figure 1G).

### 2.2. SFN Ameliorated IRI-Elicited Leukocyte and Platelet Recruitment

Having established a protective role of SFN in intestinal IRI, it was next aimed to test whether reduced enterocyte destruction at the epithelium was mediated by reduced inflammatory cell recruitment at the endothelium. Thus, multicolor IVM was used to study IRI-elicited leukocyte and platelet recruitment in postischemic intestinal postcapillary venules. It was found that intestinal IRI provoked a robust inflammatory response, associated with heightened cellular trafficking at the endothelial border, which peaked after 2 h reperfusion and then decreased over time. Most important, SFN treatment reduced this IRI-elicited cellular traffic (Figure 2A,B). It was then further aimed to dissect this result into systemic and platelet specific effects of SFN. Thus, control animal were compared to systemically SFN treated mice as well as to mice transplanted with SFN-treated platelets after 2 h, 8 h or 24 h reperfusion, respectively.

When analyzing leukocyte trafficking in untreated control mice, 30 min of intestinal IRI induced heightened leucocyte rolling and adherence, which was most prominent after 8 h (rolling) and 2 h (adherence) reperfusion. When mice were systemically treated with SFN, leucocyte rolling was significantly reduced at 2 h and 8 h reperfusion (Figure 2C). Since SFN is known to effect both leucocytes and platelets and given the fact that leucocyte recruitment depends on platelet function, it was next aimed to determine whether SFN treated platelets would alter leukocyte rolling. Thus, platelet donor mice were treated with SFN and these platelets were then exogenously labeled and adoptively transferred into untreated recipient mice, which had been exposed to intestinal IRI. When analyzing this pure platelet derived effect on IRI-elicited leucocyte recruitment, it was found that transfer of SFN treated platelets significantly reduced leucocyte rolling at 2 h and 8 h reperfusion (Figure 2C). After 24 h reperfusion, the inflammatory state was markedly decreased, and no differences in leucocyte rolling were found between groups, irrespective of the treatment modality. Next, involvement of SFN in firm leukocyte adhesion was investigated. Control mice showed significantly increased numbers of adherent leukocytes compared to systemically SFN treated mice after 2 h reperfusion (Figure 2D). At the same time-point a marked reduction between control mice (1713 ± 240.8 cells/mm^2^) and mice transplanted with SFN-treated platelets (1048 ± 204.3 cells/mm^2^) was noted. However, this was only marginally significant (*p* = 0.07) (Figure 2D). After 24 h reperfusion, both groups (systemic as well as platelet specific SFN treatment) displayed a significant reduction in firm leucocyte adhesion (Figure 2D).

Having established a protective role of systemic SFN treatment on leucocyte recruitment and having demonstrated a codependence of platelets and leukocyte adhesion in intestinal IRI, it was then thought to focus on platelet interactions with the vascular endothelium in postischemic intestinal postcapillary venules. Systemic or platelet specific SFN treatment had no effect on early platelet rolling after 2 h reperfusion (Figure 2E). However, a significant reduction was noted when comparing control mice to the platelet specific SFN group at 8 h and 24 h reperfusion (Figure 2E). For systemically SFN treated mice a trend towards reduced platelet rolling at 8 h and 24 h reperfusion was noted, however, this did not reach statistical significance. Surprisingly, no differences were found in firm platelet adhesion with either SFN treatment group (Figure 2F).

Since SFN had the most prominent effect on leucocyte and platelet rolling, it was lastly examined whether SFN would also effect rolling interactions. It was found that rolling leukocyte-platelet aggregate formation was significantly decreased in mice transplanted with SFN-treated platelets after 2 h and 8 h reperfusion (Figure 2G). In addition, the same trend was noted for systemically SFN treated animals, while this did not reach statistical significance.

### 2.3. SFN Treatment Decreased IRI-Elicited Neutrophil Infiltration in Intestinal Tissue without Altering Expression of Endothelial Adhesion Molecules

Having demonstrated a protective role of SFN in ameliorating immune cell mediated enterocyte injury and reduced cellular trafficking following SFN treatment, it was then investigated whether SFN would reduce IRI-elicited intestinal neutrophil burden. Intestinal neutrophil infiltration was evaluated by measuring MPO activity. MPO activity was found to reach a peak rapidly after 2 h reperfusion, where samples showed comparable results with or without SFN pretreatment (Figure 3A). After 8 h reperfusion MPO activity and thus neutrophil infiltration in intestinal tissue was significantly reduced by SFN treatment compared to the control group (Figure 3A). To further test whether effects on leukocyte and platelet recruitment following systemic SFN treatment were due to a SFN-induced mechanism of adhesion molecule upregulation on intestinal endothelial cells, mRNA levels for ICAM-1 (Figure 3B) and VCAM-1 (Figure 3C) were analyzed in intestinal tissue samples. Over the 24 h course of reperfusion, no statistically significant changes were noted between control and SFN treated animals.

### 2.4. SFN Reduced IRI-Elicited Ultrastructural Damage in the Intestine Protect Intestinal Barrier Integrity

Since it was hypothesized that SFN would reduce immune-mediated enterocyte destruction and thus subsequently preserve the epithelial lining in the small bowel following IRI, a special focus was the evaluation of epithelial morphology and function. First, transmission electron microscopy was used to evaluate IRI-elicited ultrastructural damage. General evaluation (i.e., general condition of the epithelium) and evaluation focusing on mitochondria and cytoplasm presented rising lesion amongst the whole period of 24 h reperfusion (Figure 4A,C,E), whereas evaluation based on microvilli and ER showed changes with no obvious trends in groups of both treatments (Figure 4B,D,F). However, a detailed and representative analysis of both groups revealed a SFN-induced amelioration of ultrastructural alterations. After 2 h of reperfusion, epithelial cells in control animals exhibited altered mitochondria and an enlarged endoplasmic reticulum, while SNF treated samples exhibited an intact morphology and organization of epithelial cells (Figure 5A,B). After 8 h of reperfusion, control animals were characterized by alterations of the brush border with a reduced number and length of the microvilli. One particular finding was that SFN treatment preserved the electron-dense tips of the microvilli, with actin filaments running to the terminal web without SFN treatment. Moreover, the terminal web of the microvilli was disorganized with a more diffuse appearance. In addition, cell-cell contacts between neighboring cells are weaker and more diffuse in control compared to SFN-treated animals. Furthermore, number of vesicles was higher, and the endoplasmic reticulum was more expanded in control samples, whereas number and structural integrity of mitochondria was not obviously altered with or without SFN treatment (Figure 5C,D). After 24 h of reperfusion, epithelial cells of control animals still displayed severe injury evident by alterations of mitochondria, endoplasmic reticulum and brush border, while SFN treated animals showed a clear structural integrity of epithelial cells (Figure 5E,F).

### 2.5. SFN Treated Mice were Less Prone to Altered Mucosal Permeability of the Small Bowel

Next, a functional assay was employed to test whether SFN would protect intestinal barrier function and block OVA translocation. Thus, to assess mucosal permeability of the small bowel following IRI, control and SFN treated animals received OVA via gavage one hour before the respective end of reperfusion. Following a peak at 2 h of reperfusion, levels of serum OVA declined over the 24 h course of reperfusion. A trend for a lower serum OVA concentration following SFN treatment was noted at all indicated time-points; however, this did not reach statistical significance (Figure 6).

## 3. Discussion

This study is the first to show protective effects of SFN in intestinal IRI. The data presented here demonstrate that SFN reduces IRI-induced leukocyte adhesion and infiltration to intestinal tissue, which subsequently leads to decreased histological damage and potentially improved intestinal barrier function. Thus, these results indicate that SFN could be a potential therapeutic strategy for intestinal IRI. While SFN represents a promising cytoprotective compound especially due to its well characterized anti-inflammatory effects [23,24,25], little is known about its role on cellular trafficking and there remains to be a gap in our current knowledge regarding its role in intestinal IRI. The present study was conducted to characterize potential protective effects of SFN treatment in a murine model of intestinal IRI. It was demonstrated that SFN elicits a protective role (evident by reduced enterocyte destruction) and that SFN treated mice were less susceptible to intestinal IRI.

The cytoprotective property of SFN is well known and has been examined in several tissues [15,16,26,27,28,29]. Mechanistically, SFN interacts with transcription factor NF-E2-related factor-2 (Nrf2), which, in a next step, activates antioxidant response element (ARE) genes [17,18,19]. The Nrf2/ARE transcriptional pathway plays an important role in protection against IRI through regulation of antioxidants and detoxification enzymes [28]. Supplementary to the well investigated antioxidant impact of SFN, this study could show additional anti-inflammatory properties, mediated by reduced leucocyte platelet interactions, resulting in protection of the intestine epithelium established by reduced inflammatory cell recruitment. Systemic SFN treatment resulted in decreased leukocyte recruitment with a most prominent effect on leukocyte and platelet rolling. Moreover, a codependence of platelets and leukocyte adhesion in intestinal IRI and decreased formation of rolling leukocyte-platelet aggregates following SFN treatment of platelets was demonstrated.

Studies investigating protective properties of SFN in IRI have mainly focused on its anti-oxidative impact, mediated by Nrf2-dependent phase enzymes which were confirmed to participate in adaptive and protective responses to oxidative stress [30]. In addition, tissue protection was previously demonstrated by reduced neutrophil infiltration and amended inflammation [31]. Thus, a positive anti-oxidative and anti-inflammatory property was verified in models of brain, heart, kidney and liver IRI [16,28,30,31,32]. The study presented herein is the first study focusing on anti-inflammatory effects of SFN concentrating on inflammatory in vivo cell trafficking utilizing a murine model of intestinal IRI. One of inherent disadvantages of the chosen species and model is the known rapid restitutional process after an early inflammatory response. Resolution of inflammation leads to a rapid cessation of immune cell recruitment, which holds also true for the here obtained data. However, especially interruptions of epithelial barrier function have been described to be present at later time points [33] and correlated with leukocyte adherence and emigration [34]. This is in line with the only marginal decrease in altered mucosal permeability as well as remaining evidence for an ultrastructural damage to the epithelium observed after 24 h in this study.

Moreover, early measurement of tissue neutrophils resulted in comparable results for control and SFN group indicating a potential discrepancy between results of intravital microscopy and MPO probably due to differences in the time frame between rolling and completeness of diapedesis into the tissue.

Leukocytes are generally recognized as key mediators of intestinal tissue injury associated with IRI, however considerably less is known about the role platelets are playing in intestinal IRI. Mounting evidence suggests a crucial role for platelets in mediating leukocyte adhesion in the post-ischemic microvasculature and as such may be crucially involved in the initiation and perpetuation of intestinal inflammation following ischemic insults. Activated platelets can bind to venular endothelial cells and create a P-selectin rich platform onto which leukocytes can roll and adhere. With respect to effects of platelets in intestinal IRI, it has been shown that neutralization of platelet-associated adhesion molecules GPIIb/IIIa or P-selectin elicited a reduction in platelet adhesion, coupled with a reduction in leukocyte adhesion and emigration in models of intestinal IRI [11,35,36,37].

Based on the results obtained in the present study, a co-dependency of platelet activation and subsequent leukocytes recruitment was confirmed via intravital microscopy in a murine model of intestinal IRI. Moreover, SFN acted as a potent pharmacological agent which not only decreased rolling leukocyte-platelet aggregate formation but also showed most prominent effects on diminishing leukocyte and platelet rolling. Systemic application of SFN altered leukocyte recruitment by reducing IRI-elicited cellular traffic and intestinal neutrophil burden.

While leukocyte recruitment is an important aspect in inflammatory response, excessive accumulation during I/R contributes to pathological conditions [38,39]. Recent studies have shown the ability of SFN to affect cells of the innate immune system including neutrophils and platelets that are known as the first line defense following injury [20,40]. For leukocytes, it has been shown that SFN is able to alter leukocyte adhesion. Moreover, it is suggested to play a role in the regulation of pro-inflammatory cytokines and, through the role as Nrf2-agonist, required for optimal migration of neutrophils [40]. In accordance with the here presented data, a study of LPS-induced inflammation in the brain could show a decrease in leukocyte adherence as well as reduced MPO levels after pre-treatment with SFN [41]. In addition, protective actions of SFN were demonstrated by down regulation of E-selectin and VCAM-1 in endothelial cells as well a decrease in ROS production in SFN treated human neutrophils. More recently, an additional mechanism of action for SFN influencing platelets was discovered. SFN-pretreatment resulted in a reduced platelet activation quantified by decreased levels of P-selectin and integrin αIIbβ3. Moreover, platelet aggregation as well as adhesion was altered under inflammatory conditions [22]. While this data demonstrates that SFN can alter platelets, vascular endothelial cells as well as leukocytes in the cascade of inflammatory-induced immune cell activation and recruitment, our current in vivo data suggest a more relevant effect on platelets and leukocytes since no change in the expression of adhesion molecules ICAM-1 and VCAM-1 were noted in systemically treated tissue samples, while platelet specific SFN treatment reduced leukocyte and platelet rolling as well as rolling interactions of both cell types. However, to fully discriminate between a more dominant effect of SFN on platelets or leukocytes in intestinal IRI, further studies are needed in which SFN treated leukocytes are transfused into leukocyte depleted animals.

Although the current data clearly shows a reduction of leukocyte and platelet recruitment as well ameliorated intestinal neutrophil burden, other cell types (such as monocytes or T cells) are also critically linked to IRI [2]. While the current study focuses on neutrophils, well-described effects of SFN on other cell lines of the innate [20,42] and adaptive [43] immune system suggests that the reduced intestinal inflammatory phenotype following SFN treatment might at least be in part elicited by reduced monocyte or T cell recruitment. Further studies are needed to more precisely dissect these mechanisms.

In summary, the present study shows for the first time protective effects of SFN in a murine model of intestinal IRI with a specific focus on cellular trafficking. Moreover, these results further highlight the importance of platelets in mediating leukocyte adhesion in the post-ischemic microvasculature and their crucially involvement in initiation and perpetuation of intestinal inflammation following IRI. The results obtained herein, could confirm anti-inflammatory actions of SFN regarding leukocyte and platelet recruitment. The therapeutic benefit of pretreatment of platelets with SFN and the molecular mechanism of action resulting in reduced leukocyte activation will require further investigation. Especially small bowel transplantation with its inherent acute ischemia reperfusion injury [44] as well as common inflammatory rejection episodes, represents a potential clinical application for the uses of SFN as possible therapeutic agent. While, SFN has been successfully tested in humans [45], there is no clinical data available in the field of human transplant. However, there are two reports demonstrating protective effects on renal and cardiac grafts [31,32]. Hence, the present findings are promising in that SFN provides a working point for ameliorating intestinal IRI through modulation of leukocyte and platelet activation.

## 4. Materials and Methods

### 4.1. Ethical Approval

All experiments involving animals were performed in accordance with the German Animal Welfare Law and beforehand approved by the administrative authority of North Rhine-Westphalia (Landesamt für Natur, Umwelt und Verbraucherschutz Nordrhein-Westfalen, reference number 84-02.04.2017.A152. date of approval 17 08 2017). Reporting of animal research within this study was conducted according to the ARRIVE guidelines.

### 4.2. Animals

Eight to eleven weeks old male wild type C57BL/6J mice (Charles River Laboratories, Sulzfeld, Germany) were used. All animals had at least one week prior to surgery as acclimatization time, allowing free access to autoclaved tap water and regular chew food *ad libitum*. Housing environment was kept at automated 12 h light/dark cycles with room temperature of 24 ± 2 °C.

### 4.3. Experimental Groups and Treatment

Animals were randomly assigned to one of the following three experimental groups: (1) control (CON), (2) systemic SFN treatment (SFN-S) and (3) platelet specific SFN (SFN-P) treatment. Control animals in group 1 received an intraperitoneal (ip) injection of 10 mL/kg body weight (BW) dissolvent (mix of corn oil and sterilized phosphate buffered saline (PBS), ratio of 1:9)) 24 h prior induction of intestinal IRI. Animals undergoing systemic SFN treatment in group 2 received an ip injection of SFN (5 mg/kg BW, L-Sulphoraphane, Sigma-Aldrich, Darmstadt, Germany, dissolved in a mix of 10% corn oil and 90% PBS) 24 h prior induction of intestinal IRI. For platelet specific SFN treatment in group 3, platelet donor mice received 5 mg/kg BW SFN (Sigma-Aldrich) ip 24 h prior to platelet donation. SFN-P animals were only used for intravital microscopy and are not included in datasets for any downstream assay (for further details see Appendix A).

### 4.4. In Vivo Model of Intestinal IRI

A temporary vascular occlusion of the superior mesenteric artery (SMA) was used to establish intestinal IRI. Thirty minutes prior to surgery, analgesia treatment was started by subcutaneous injection of buprenorphine (Indivior UK Limited, Slough, UK) and continued every eight hours (0.1 mg/kg BW) during follow up. Next, animals were anesthetized by inhalation of 1.5 vol.% isoflurane (Forene 100% Abbott, Wiesbaden, Germany). After surgical tolerance was reached, the abdominal wall was opened via a midline incision and the SMA was carefully mobilized. A microsurgical bulldog clamp was used for temporary artery occlusion and induction of ischemia. After 30 min of ischemia, the clamp was carefully removed, and the small bowel was reperfused. During the procedure, body temperature was kept at 37 °C with a heating pad and the small bowel was protected with warm and moist gauze. After surgery the abdominal incision was sutured in two-layer fashion and animals were returned back to housing boxes with free access to food and water during reperfusion span of 2, 8 or 24 h, respectively.

### 4.5. Ex Vivo Platelet Preparation

Approximately 0.9 mL of blood was harvested from donor mice via a carotid artery catheter into a syringe prefilled with 0.1 mL acid-citrate-dextrose solution (ACD, Sigma-Aldrich). Platelet rich plasma was yielded via sequential centrifugations at 1400 rpm for 8 min and 3 min. Next, platelet pellet was obtained by centrifuging platelet rich plasma supernatant of 3000 rpm for 10 min and resuspending the obtained pellet with 500 µL sterilized PBS. Platelets were incubated with fluorochrome carboxyfluorescein diacetate succinimidyl ester (CFSE, Thermo Fisher Scientific, Waltham, MA, USA) at room temperature for 10 min followed by centrifugation of 3000 rpm for 10 min. Simultaneously, platelet quantification was performed with a hemocytometer. After centrifugation, fluorescently labelled platelets were again suspended with PBS and 100 × 10^6^ platelets were infused into recipient mice via a jugular vein catheter 5 min before intravital microscope examination. Recipient mice were not rendered thrombocytopenic before infusion of donor platelets.

### 4.6. In Vivo Leukocyte Staining and Fluorescence Intravital Microscopy (IVM)

As previously described, rhodamine 6G (100 μL, 20 mg/mL, Sigma-Aldrich) was used for fluorescence in vivo leukocyte labelling 5 min prior to IVM. Next, mice underwent relaparotomy and a proximal intestinal segment was fixed for in vivo analysis of interactions between platelets, leukocytes and vascular endothelial cells in postischemic intestinal postcapillary venules (pcvs). Animal’s body temperature was maintained at 37 °C with a heating pad and exposed intestine was kept wet by periodic administration of warm saline. A multicolor intravital fluorescence microscope (BX51WI, Olympus Life Science Europa GmbH, Hamburg, Germany) equipped with a 20× water-immersion objective (Zeiss, Oberkochen, Germany) and a wavelength switching illumination system (Lambda DG-4, Novato, Canada) was applied. Digital visualization and offline recording of individual vessels and fluorescently labelled cells were performed using an emission splitting system in combination with a microscope camera (Cascade II 512, Photometrics, Tucson, AZ, USA). In all animals, pcvs (15–30 µm of diameters) in three randomly selected and non-overlapping microscopic fields were scanned and recorded for 90 s with one picture every 50 ms.

### 4.7. Offline Video Analysis

In offline video analysis video analysis, the software VisiView (Version 3, Visitron Systems GmbH, Puchheim, Germany) was used. Platelets and leukocytes were quantified by respective characteristics and in accordance to rheological properties classified as (a) free flowing (no intercellular or endothelial contact), (b) rolling (slower than centerline blood flow) and (c) adherent (sticking to endothelium for more than 30 s) [46]. Rolling velocity was expressed as number of rolling cells per second per millimeter of vessel diameter (cells/s/mm), whereas adhesion was presented as number of sticking cells per square millimeter of vessel surface calculated by diameter and length (cells/mm^2^). Interactions between platelets and leukocytes, platelets and endothelial cells, leukocytes and endothelial cells as well as all three types of cells were documented independently.

### 4.8. Intestinal Histology

After isolation and dissection of small bowel, ileum was cut open longitudinally and prepared according to the “Swiss roll” technique [47,48]. All operations were conducted on ice. Tissue samples were fixed in 4% formaldehyde solution (Otto Fischar, Saarbruecken, Germany) for 24 h and embedded in paraffin. Intestinal histology of hematoxylin and eosin stained sections was assessed by a pathologist in blinded fashion in accordance to Park/Chiu score system, as previously described [49,50].

### 4.9. Electron Microscope

Small pieces of ileum were fixed in 2% (*v/v*) formaldehyde and 2.5% (*v/v*) glutaraldehyde in 100 mM cacodylate buffer, pH 7.4, at 4 °C. After washing in PBS, specimens were post-fixed in 0.5% (*v/v*) osmiumtetroxide and 1% (*w/v*) potassium hexacyanoferrate (III) in 0.1 M cacodylate buffer for 2 h at 4 °C followed by intense washing with distilled water. After dehydration in an ascending ethanol series from 30 to 100% ethanol, specimens were incubated two times in propylenoxide each for 15 min. Next, tissue pieces were embedded in Epon using flat embedding molds. Ultrathin sections were cut with an ultramicrotome, collected on copper grids, and negatively stained with 2% uranyl acetate for 15 min. Electron micrographs were taken at a Phillips EM-410 electron microscope using imaging plates (Ditabis, Pforzheim, Germany). The ultrastructure of intestinal epithelia was examined in a blinded fashion. Evaluation was performed focusing on microvilli and organelles integrity including the endoplasmic reticulum (ER) and mitochondria swelling, electron density, nucleus integrity and the general condition of the cytoplasm.

### 4.10. Intestinal Tissue Myeloperoxidase (MPO) Activity

MPO activity was measured using the O-dianisidine Myeloperoxidase Assay as previously described [51]. Samples procured from the jejunum were snap frozen in liquid nitrogen and stored at −80°C. To determine MPO activity, tissue was homogenized and subsequently sonicated three times for 15 s in a ratio of 0.1 g/1 mL in 0.5% (*w/v*) hexadecyltrimethylammonium bromide (HETAB >98%, Sigma-Aldrich) dissolved in 50 mM potassium phosphate buffer (KPi, pH 6.0). Samples were centrifuged at 12,000 rpm for 10 min at 4 °C and supernatant mixed with 10 mL chloroform (EMD Millipore Corporation, Darmstadt, Germany) and vortexed three times for 10 s for lipid extraction. Supernatants were again centrifuged at 12,000 rpm for 5 min at 4 °C. Upper layer was collected for later reaction. All operations were performed on ice. The reaction system consisting of 2.81 mL of 50 mM KPi (pH 6.0), 30 µL of 20 mg/mL o-Dianisidine dihydrochloride (OD >95%, Sigma-Aldrich) and 30 µL of 20 mM hydrogen peroxide was mixed with 100 µL of sample obtained as described above. OD-blank controls were conducted with 30 µL of distilled water replacing OD. Sample-blank control was set with 100 µL of distilled water as substitute of sample. After 10 min of incubation at room temperature, reaction was stopped by adding 30 µL of 20 mg/mL sodium azide (Sigma-Aldrich). Absorbance of reaction mix was tested with a S11 spectrophotometer (Biochrom GmbH, Berlin, Germany) at wavelength of 460 nm. Results were calibrated by subtracting the larger absorbance amongst the OD-blank and sample-blank controls from the one of each sample. Calculation of MPO activity (U/g) was conducted by dividing the calibrated absorbance by reaction time (10 min) and tissue wet weight in gram.

### 4.11. Intestinal Mucosal Permeability

Mice received 100 µL of Ovalbumin (OVA) solution (400 µg/mL) dissolved in sterile PBS one hour prior to the end of reperfusion. Afterwards blood was collected via puncture of the heart and drained into a micro tube (1.3 mL, Clotting Activator/Serum, SARSTEDT AG & Co., Nümbrecht, Germany). After 24 h storage at 4 °C, serum was obtained by centrifugation at 10,000 rpm for 5 min at room temperature. OVA concentration in serum was determined with a commercially available enzyme-linked immunosorbent assay kit (General Ovalbumin (OVA) ELISA Kit, MyBioSource, San Diego, CA, USA) according to manufacturer’s instruction.

### 4.12. Quantitative Polymerase Chain Reaction (qPCR)

Intestinal tissue samples were collected after ischemia and reperfusion and temporarily stored in a stabilization solution (RNAlater™ Stabilization Solution, Thermo Fisher Scientific) over night at 4 °C. Samples were then moved to a cryo tube (CryoPure, SARSTEDT AG & Co., Nümbrecht, Germany) for long-term storage at -80 °C. RNA isolation from tissue samples was performed with an RNeasy Mini Kit (QIAGEN GmbH, Hilden, Germany) according to manufacturer’s instruction. Purified RNA was then reversely transcribed into cDNA with a QuantiTect Reverse Transcription Kit (QIAGEN GmbH). For the next step of qPCR, primers of target genes ICAM-1 (QT00155078), VCAM-1 (QT00128793) as well as glyceraldehyde 3-phosphate dehydrogenase (GAPDH, QT01658692) as housekeeping gene were purchased from QIAGEN GmbH (QuantiTect^®^ Primer Assay, Qiagen GmbH). Reaction mix prepared with reagents from a QuantiTect SYBR Green PCR Kit (QIAGEN GmbH) was applied to establish the reaction system and mixed with cDNA and primers of different target genes, respectively. The qPCR of target genes was performed in a CFX384 real-time PCR detection system (BIO-RAD, Feldkirchen, Germany) with amplification results documented and analyzed with a programme Bio-Rad CFX Manager (BIO-RAD). Results were calculated by using the 2^−ΔΔC^T method and presented as relative gene expression.

### 4.13. Statistics

Data analysis was performed with GraphPad Prism (GraphPad Software, Version 5.0, San Diego, USA). Unpaired Student t test was applied for statistical analysis between two groups, while comparison between three groups was conducted by variance analysis and post hoc testing (one-way ANOVA with Bonferroni post hoc testing). Results are presented as mean values with standard error of mean (mean ± SEM). Differences were considered statistically significant when *p* value < 0.05.

## Figures and Tables

**Figure 1 ijms-21-05189-f001:**
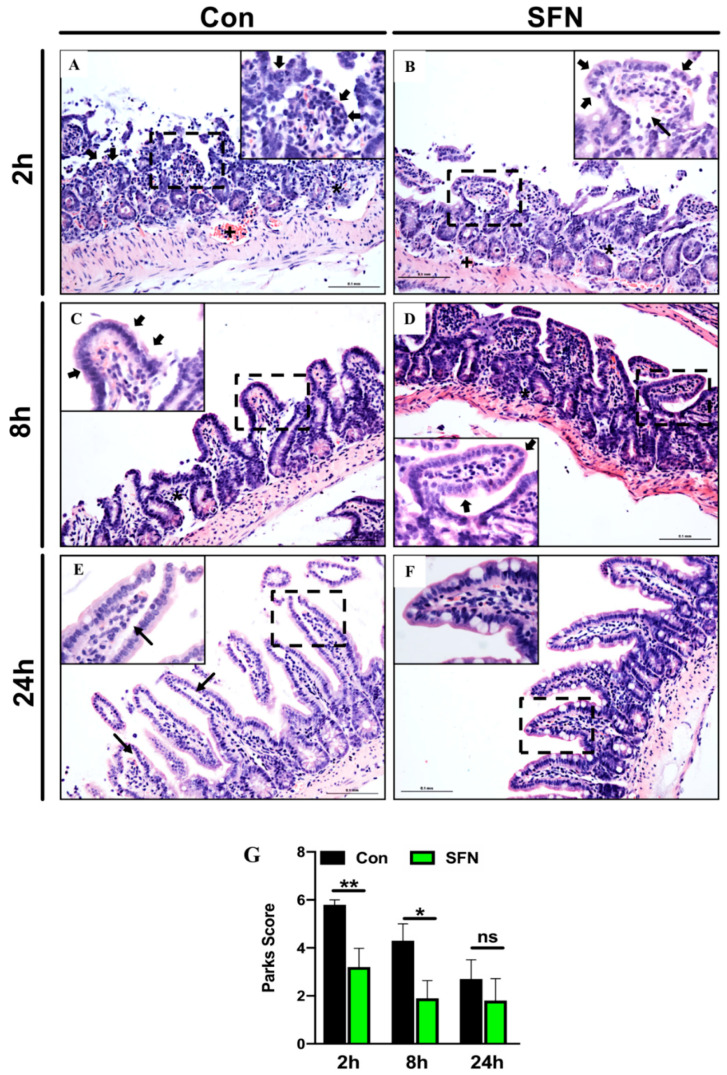
(**A**–**F**) Representative histopathologic images of hematoxylin and eosin (H&E) stained cross sections showing IRI-induced morphological damage in the murine intestine of control (CON) and SFN-treated animals (SFN). (**G**) Park/Chiu score comparing comparing control (CON) and SFN-treated animals (SFN) after 2 h, 8 h and 24 h of reperfusion. Overview images were taken at × 20 magnification, insets (dashed square in the overview image) at × 40 magnification, scale bar 0.1 mm. Black asterix showing submucosal mixed inflammatory cell infiltrate, black cross showing dilated capillaries, filled black arrows indicate epithelium lining at the top of villi and small black arrows indicate extension of the subepithelial space. All data are presented as mean values ± SEM. *n* = 10 mice per group. Significance is indicated by the following symbols: * = *p* < 0.05, ** = *p* < 0.01 versus Con, ns = not significant.

**Figure 2 ijms-21-05189-f002:**
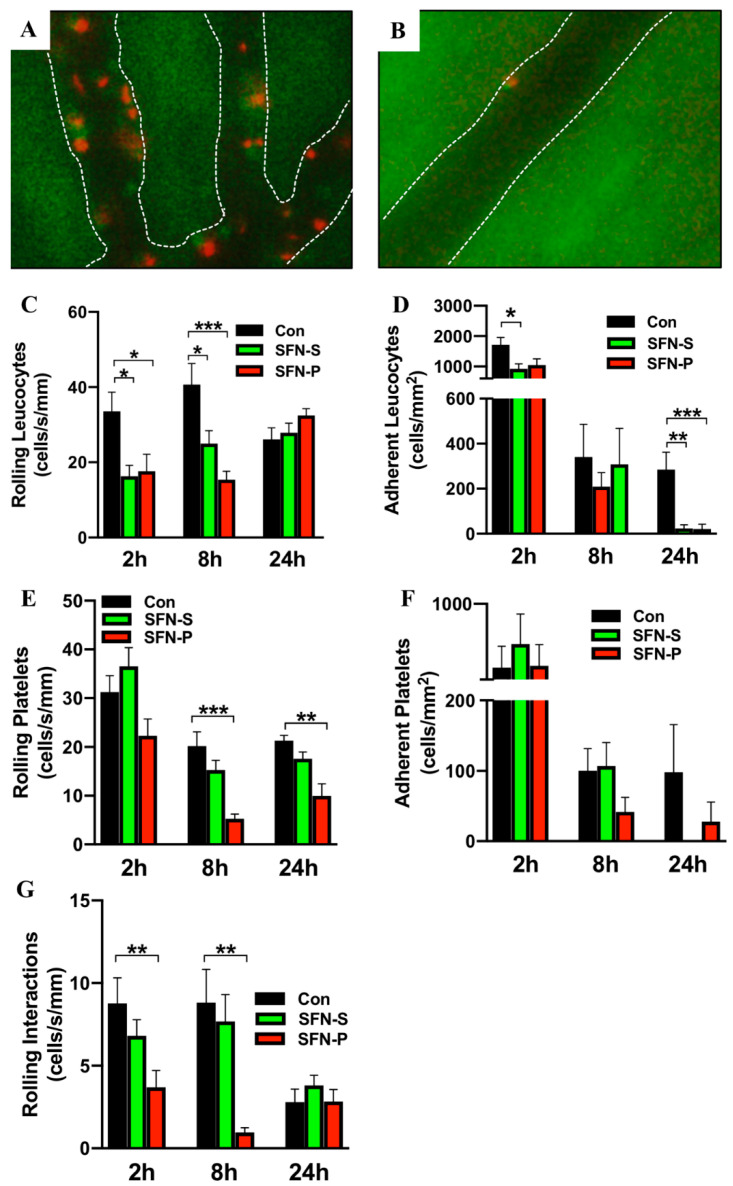
Representative intravital microscopy (IVM) images of leukocytes (red) and platelets (green) in postischemic intestinal postcapillary venules of (**A**) control and (**B**) SFN-treated mice, white lines mark the vascular boarder. (**C**–**F**) Leukocyte and platelet trafficking was analyzed via rolling (**C,E**) and adherence (**D,F**) in postischemic intestinal postcapillary venules of control (CON), systemically treated (SFN-S) and platelet treated (SFN-P) mice with 2 h, 8 h and 24 h reperfusion. (**G**) Rolling interactions of leukocytes and platelets in control (CON), systemically treated (SFN-S) and platelet treated (SFN-P) mice with 2 h, 8 h and 24 h reperfusion. All data are presented as mean values ± SEM, n= 9-11 animals per group. Significance is indicated by the following symbols: * = *p* < 0.05, ** = *p* < 0.01, *** = *p* < 0.001 versus Con.

**Figure 3 ijms-21-05189-f003:**
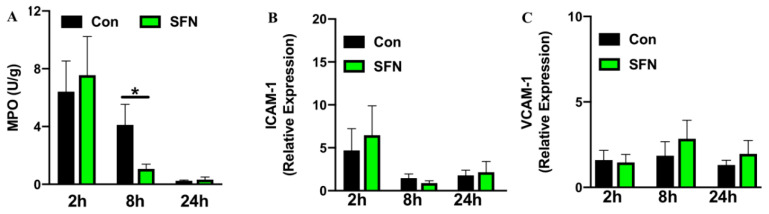
Intestinal neutrophil infiltration measured by MPO activity (**A**) and expression of ICAM-1 (**B**) and VCAM-1 (**C**) determined by qPCR and presented as relative gene expression. All assays were conducted in tissue samples (ileum) of control (CON) and SFN-treated (SFN) animals with 2 h, 8 h and 24 h of reperfusion. All data are presented as mean values ± SEM, n = 8–10 animals per group. Significance is indicated by the following symbol: * = *p* < 0.05 versus Con.

**Figure 4 ijms-21-05189-f004:**
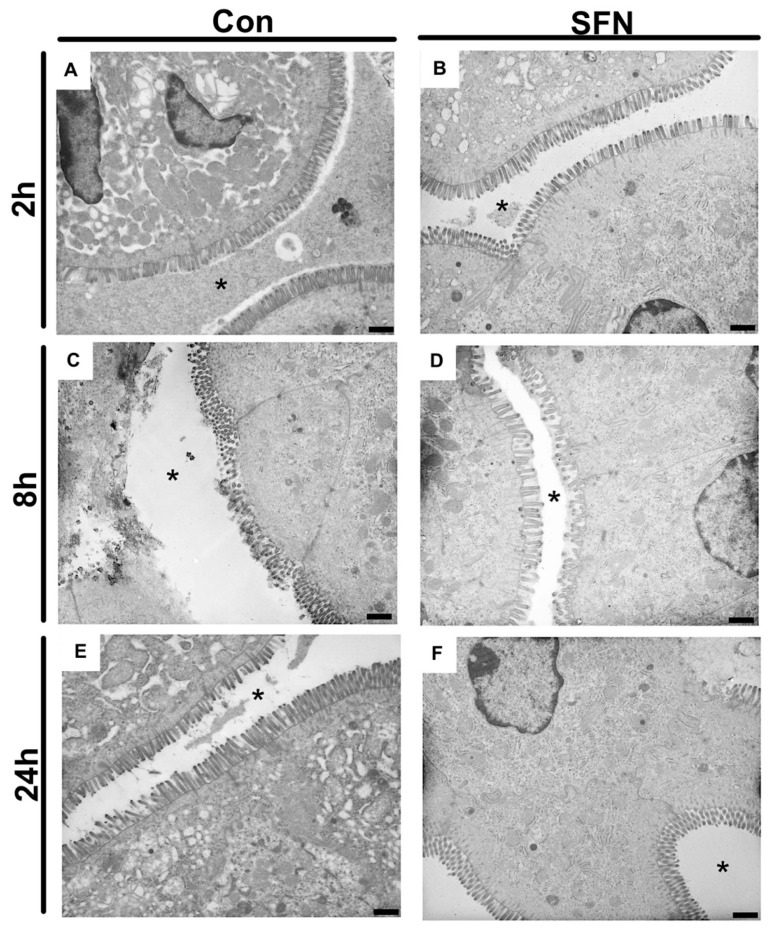
Representative electron micrographs of controls (**A**,**C**,**E**) and treated samples (**B**,**D**,**F**) after 2 h, 8 h and 24 h of reperfusion. * = intestinal lumen; n = nucleus; mv = microvilli. Images are representative of n = 10 analyzed animals per group and timepoint. All bars: 1 µm.

**Figure 5 ijms-21-05189-f005:**
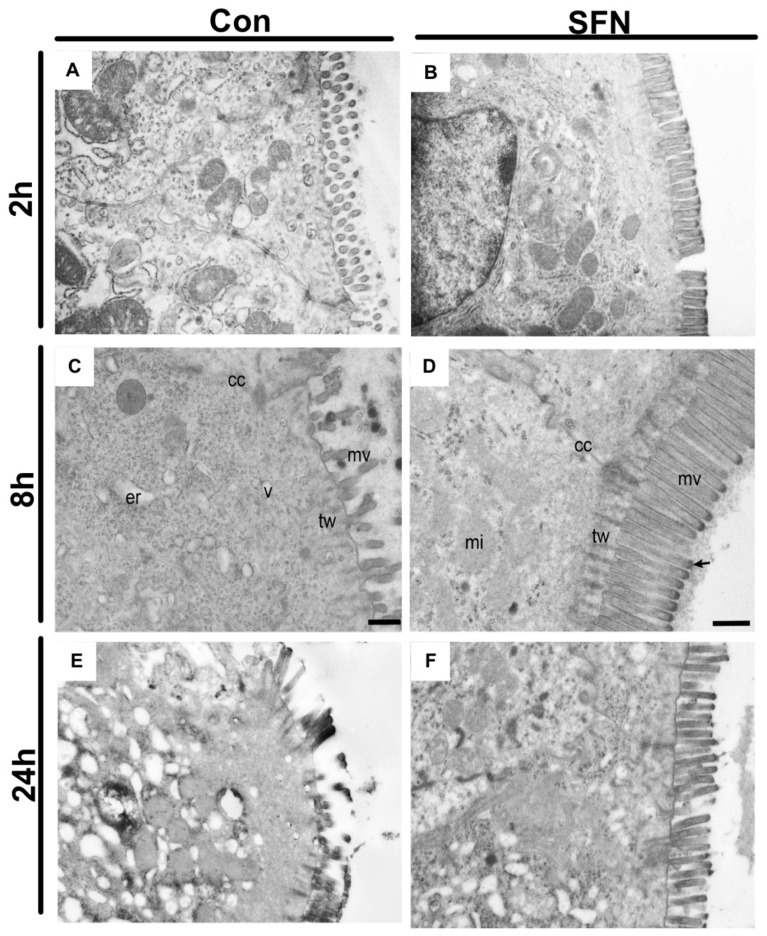
Representative electron micrographs of control (**A**,**C**,**D**) and SFN treated (**B**,**D**,**F**) samples after 2, 8 and 24 h of reperfusion. (**A**) In controls after 2 h reperfusion, epithelial cells exhibit altered mitochondria and an enlarged endoplasmic reticulum, while SNF treated samples exhibits an intact morphology and organization of epithelial cells (**B**). (**C**) Epithelial cell with an altered brush border with less microvilli and a disorganized terminal web. Numerous vesicles and an enlarged endoplasmic reticulum are clearly visible while cell-cell contacts are disrupted. (**D**) Epithelial cells connected by intact cell-cell contacts. Ultrastructure of the brush border and terminal web region with highly organized microvilli. Notice the actin filaments that descend from the electron dense tip (arrow) of the microvillus into the underlying terminal web. (**E**) A strongly affected brush border is clearly visible after in control samples after 24 h reperfusion, which is absent in SFN treated animals, showing an intact morphology and organization of epithelial cells (**F**). Images are representative of n = 10 analyzed animals per group and timepoint. cc = cell-cell contact; mi = mitochondria; mv = microvilli; tw = terminal web; er = endoplasmic reticulum. Scale bar for all images: 500 nm.

**Figure 6 ijms-21-05189-f006:**
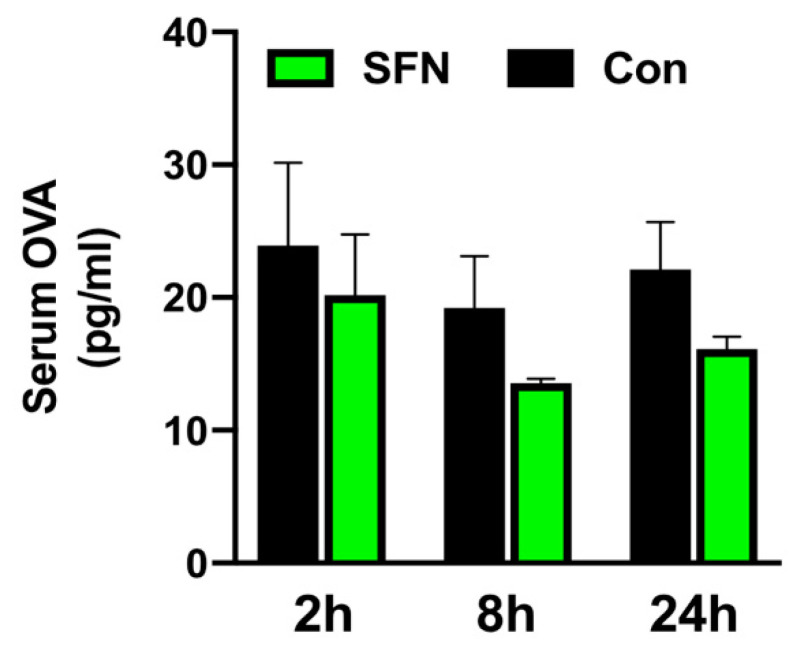
Protection of intestinal barrier function was analyzed by OVA translocation to the serum in control (CON) and SFN-treated (SFN) animals after 2, 8 and 24 h of reperfusion. All data are presented as mean values ± SEM. n = 6 animals per group.

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
