# Peer review of "Sulforaphane Elicits Protective Effects in Intestinal Ischemia Reperfusion Injury"

_ijms, 2020, doi:10.3390/ijms21155189_

Round 1

Reviewer 1 Report

The authors have investigated the role of SFN in reducing intestinal IRI in a mouse model of disease.  The authors present interesting observational data regarding their model which is of value to the study of IRI.  However, their data would be strengthened by a few experiments to support their hypothesis that SFN is protective in intestinal IRI by limiting leukocyte and platelet function/recruitment.

  1. The authors should deplete leukocytes with adoptive transfer of treated versus untreated SFN leukocytes prior to intestinal IRI to demonstrate that SFN has a leukocyte specific effect
  2. Can the author's clarify if they depleted platelets prior to infusion of SFN treated platelets or simply treated platelets ex vivo prior to infusion
  3. Do the authors have data regarding expression of P or L selectin on vascular endothelium vis a vis SFN treatment as that appears to part of their hypothesis. This appears to be an important piece of data given the hypothesis that SFN affects leukocyte and platelet rolling and attachment.
  4. The authors should be able to identify which leukocyte populations are reduced at each time point. (i.e. early neutrophils, then monocytes, etc.)  This would be important data to understand the functional effects of SFN.
  5. Minor point: could the authors add in figure 1 arrows to indicate areas of difference between the h and e sections (particularly the areas measured for the IRI score). this would help illuminate differences for readers.

Reviewer 2 Report

Dear authors,

The authors report for the first time the protective effect of sulforaphane (SFN) in decreasing the intestinal ischemia reperfusion injury, through the decrease of the leukocyte rolling and adhesion. This study was conducted on mice. A single injection of SFN decreased the level of Parks injury, the level of rolling and adhesion leukocyte and the level of Myeloperoxidase, up to 8h. In addition, the SFN protected the intestinal barrier integrity to the IRI. This paper is important because it is the first study reporting the protective effect of the sulforaphane on intestinal ischemia reperfusion.

I have questions about the manuscript:

Major:

  • Why the authors do not report data for SFN-N and SFN-P for all the figures? What is the difference between SFN and SFN-N/SFN-P? Are control group mice had IR, in absence of treatment?
  • How many mice were used in total and per group?
  • What is the n for each figure?
  • In the discussion (line 220 to 223), the authors mentioned that sulforaphane have a potential therapeutic to decrease IRI. Sulforaphane shows a protective effect up to 8h, but at 24h there is no difference with the control group.
  • In figure 5, did the authors noticed that the changes/unchanges of the vesicles and the endoplasmic reticulum at 2h and 24h? Data from 2h and 24h should be provided for comparison.
  • How long intestine can be preserved during an ischemia-reperfusion? Does the 2-8h protection by the sulforaphane enough to protect the intestine from injuries? This subject should be comment on the discussion part of the manuscript.

Minor:

  • Latin words must be written in italic (in vivo…)

Reviewer 3 Report

Dear Madame, dear Sir, first I would like to congratulate you for your fine work. 

However, I do have some suggestions to further improve your work. I do think that the reader -who is not so familiar with the methodds as well as the Intravital Microscopy Method- would benefit a) if you implement a grapg with the timeline and interventions of the experiment. e.g. -7d Arrival of the animals and acclimatisation, 0d Anesthesia and Operation time of analgesia etc. b) a graph or photo of the IVM setup with the mouse

You measured MPO and at 2h the MPO level was a high as in control. 

You wrote in the manuscript: 

242 Studies investigating protective properties of SFN in IRI have mainly focused on its  anti-oxidative

243impact, mediated by Nrf2-dependent phase enzymes which were confirmed to

244 participate in adaptive and protective responses to oxidative stress [36].

However, when the protective effects is mainly based on the anti-oxidative propoerties and the MPO level at 2h is a high as in the control group you should adress this issue more in the discussion and speculate why the 2h level is not reduced. 

Pls check you manuscript according to the ARRIVE Guideline Checklist and state afterwards that you comply

Minor: Did you used software for the Offline video Analysis of the IVM? If yes pls report software and version

You used Grpah Pad pls report Version

Round 2

Reviewer 1 Report

Acceptable for publication

Reviewer 2 Report

Dear Authors,

I have few minors comments:

  • In Figure 6, a horizontal line attached to SFN and Con box should be removed.
  • For the references: the number of the reference is written twice (1. 1. ; 2. 2.;...). Also the year of the publication should be written after the abbreviation of the journal. (https://www.mdpi.com/journal/ijms/instructions)

Sincerely
